# Synthesis, Antimicrobial and Antioxidant Activities of 2-Isoxazoline Derivatives

**DOI:** 10.3390/molecules25184271

**Published:** 2020-09-18

**Authors:** Asma Alshamari, Mahmoud Al-Qudah, Fedaa Hamadeh, Lo’ay Al-Momani, Sultan Abu-Orabi

**Affiliations:** 1Department of Chemistry, Faculty of Science, University of Ha’il, P.O. Box. 2440, Ha’il 81451, Saudi Arabia; aasmaalshamari@gmail.com; 2Department of Chemistry, Faculty of Science, Yarmouk University, P.O. Box.566, Irbid 21163, Jordan; mqudah74@gmail.com (F.H.); abuorabi@yu.edu.jo (S.A.-O.); 3Department of Chemistry, Tafila Technical University, P.O. Box. 179, Tafila 66110, Jordan; loay1970@yahoo.com

**Keywords:** isoxazolines, nitrile oxide, 1,3-dipolar cycloaddition, antibacterial activity, antioxidant activity

## Abstract

A series of derivatives of *trans*-3-(2,4,6-trimethoxyphenyl)4,5-dihydroisoxazolo-4,5-bis[carbonyl-(4′phenyl)thiosemicarbazide (**9**) and of *trans*-3-(2,4,6-trimethoxyphenyl)-4,5-dihydro isoxazolo-4,5-bis(aroylcarbohydrazide) (**10a**–**c**) were synthesized from *trans*-3-(2,4,6-trimethoxyphenyl)-4,5-dihydro-4,5-bis(hydrazenocarbonyl)isoxazole (**8**). The structures of the compounds were elucidated by both elemental and spectral (IR, NMR, and MS) analysis. Compound **9** shows activity against some bacterial species. No antibacterial activities were observed for compounds **10a**–**c**. The antioxidant activity of the new compounds has been screened. Compound **9** showed higher antioxidant activity using the DPPH (1,1-diphenyl-2-picrylhydrazyl) and ABTS (2’-azino–bis(3-ethylbenzoline-6-sulfonic acid) diammonium salt methods.

## 1. Introduction

Isoxazolines are an important class of nitrogen- and oxygen-containing heterocycles that belong to the azoles family, and have gained great importance in the field of medicinal chemistry as anticancer agents [1,2,3,4,5,6,7,8]. They are also reported to possess good antimicrobial, analgesic, anti-inflammatory activities [4]. Several isoxazoline derivatives were generated by 1,3-dipolar cycloaddition of nitrile oxides. Abu-Orabi et al. reported that nitrile oxides (**1a** and **1b**) reacted with dialkyl maleate (**2a** and **2b**) to afford only the *cis*-cycloadducts **3a**–**d**; Scheme 1 [9,10].

In contrast, the reaction of nitrile oxides **1a** and **1b** with dialkyl fumarate (**4a** and **4b**) yielded the *trans*-isomers **5a**–**d**; Scheme 2 [9,10,11].

Similarly, the reaction of compounds **1a** and **1b** with *trans*-dibenzoylethylene (**6**) was carried out to obtain the corresponding 2-isoxazoline derivatives **7a** and **7b** in good yields, as shown in Scheme 3 [11].

The isoxazoline derivatives **5a**–**d** bearing an ester group were reacted with excess hydrazine hydrate in ethanol under reflux to give high yields of bis(hydrazinocarbonyl) derivatives **8a**–**b**; Scheme 4 [12]. Isoxazole derivatives **5a**–**d** showed antitrypanosomal activity [13].

Isoxazoline derivatives exhibit biological and pharmaceutical activities. They also have wide industrial and analytical applications [1,2,3,4,5,6,7,8,13]. These observations prompted us to study the synthesis of certain compounds that possess the isoxazoline moiety to study the antibacterial and antioxidant activities of these compounds on different bacterial strains.

## 2. Results and Discussion

Compound **9** was prepared from the reaction of *trans*-3-(2,4,6-trimethoxyphenyl)-4,5-dihydro-4,5-bis(hydrazinocarbonyl)isoxazole (**8a**) with phenylisothiocyanate in absolute ethanol at room temperature; Scheme 5.

The High resolution mass spectrometry (HRMS) for compound **9** displays a molecular ion peak at *m*/*z* = 622.16207 corresponding to the ion [C_28_H_29_N_7_O_6_S_2_-H]^−^, as expected from its calculated *m*/*z* = 622.15425. The IR spectrum of the thiosemicarbazides derivative **9** shows absorption bands in the range 3028–3363 cm^−1^, which are assigned to the N-H stretching frequency. The band at 1703 cm^−1^ is assigned to the carbonyl group, while the band in the range 1200–1256 cm^−1^ corresponds to the thiocarbonyl group. The ^1^H-NMR spectrum for compound **9** shows peaks in the range 9.75–10.52 ppm, which belong to the NH protons. These protons are deuterium exchangeable—the ten aromatic protons are detected as a multiplet in the range 7.15–7.44 ppm, and the other two phenyl protons appear as a singlet at 6.20 ppm. The protons on carbon atoms 4 and 5 of the isoxazoline ring appear as two doublets at 4.85 and 5.40 ppm, with a coupling constant of 7.8 Hz. The NH peaks were confirmed upon addition of deuterium oxide to the NMR tubes of compound **9**.

The ^13^C-NMR spectrum of compound **9** shows the carbon signal of C=O groups at 159 and 162 ppm. The signal at 180 ppm is assigned to the carbon in C=S bond—ten aromatic carbons were observed in the range 124–128 ppm. The quaternary carbons in the aromatic rings appear at 138 ppm.

### 2.1. Preparation of trans-3-(2,4,6-Trimethoxyphenyl)-4,5-dihydroisoxazolo-4,5-bis(phenylcarbohydrazide) ***10a**–**c***

Compounds **10a**–**c** were prepared from the reaction of *trans*-3-(2,4,6-trimethoxyphenyl)-4,5-dihydro-4,5-bis(hydrazinocarbonyl)isoxazole **8a** with two equivalents of aroyl chloride derivatives, as shown in Scheme 1. The dicarbohydrazides **10a**–**c** were characterized by the IR, ^1^H-NMR, ^13^C-NMR, and HR-MS.

The HRMS for compound **10a** displays a molecular ion peak at *m*/*z* = 560.17793 corresponding to the ion [C_28_ H_26_N_5_O_8_-H]^−^, as expected from its calculated *m*/*z* = 560.17814. The IR spectrum shows a broad band in the range 3008–3306 cm^−1^, which is assigned to the NH stretching frequency. The band at 1650 cm^−1^ indicates the amide carbonyl group. In compound **10c**, the band at 1535 cm^−1^ corresponds to the nitro (NO_2_) group.

It can be seen from the ^1^H-NMR spectra of compound **10a**–**c** that the three methoxy groups appear as two singlets—one at 3.76 ppm corresponds to six protons, and it belongs to the methoxy at the ortho positions of the phenyl group, while the singlet at 3.79 ppm corresponds to the protons of the methoxy group in the para position. The two protons on carbon atoms 4 and 5 of the isoxazoline ring appear as two doublets at 5.02 and 5.31 ppm, with a coupling constant of 7.8 Hz. The two protons of the aryl group appear at 6.32 ppm as a singlet. Compound **10a** has ten aromatic protons as a multiplet in the range 6.93–7.97 ppm, according to the ^1^H-NMR spectrum. The broad peak in the range 10.27–10.59 ppm is assigned to the four NH amide protons.

The ^1^H-NMR spectra of compound **10b** shows a single peak at 3.83 ppm, which corresponds to the protons of the two methoxy groups in the *meta* position on the benzoyl rings, and eight aromatic protons appear as a multiplet in the range 7.11–7.50 ppm. The broad peak in the range 10.22–10.53 ppm is assigned to the four NH amide protons, while ^1^H-NMR spectrum of compound 1**0c** shows eight aromatic protons as a multiplet in the range 7.60–8.12 ppm. Three singlets in the range 10.57–10.81 ppm are assigned to the four NH amide protons. The ^13^C-NMR spectra of the compounds **10a**–**c,** show signals at 55.3 and 55.5 ppm, which are assigned to the three methoxy carbons on the phenyl group; the signal at 90 ppm is assigned to the two CH aromatic carbons. The C=O carbons appear at 164–167 ppm. For compound **10a,** the aromatic carbons in the two benzoyl rings appear in the range 124–139 ppm, while in compound **10b**, the aromatic carbons appear in the range 112–133 ppm. The two methoxy groups attached to the benzoyl ring appear at 55.2 ppm. In compound **10c**, signals at 146 and 147 ppm are assigned to the carbons attached to the nitro group—other aromatic carbons appear in the range 124–133 ppm. The full assignments of ^13^C–NMR and ^1^H–NMR chemical shifts of compound 10b were confirmed by HMQC and HMBC. The IR, ^1^H-NMR and ^13^C-NMR spectra of all the compounds are presented in the Appendix A.

### 2.2. Antimicrobial Activity

The antibacterial activities of the newly synthesized compounds were evaluated in vitro against three Gram-positive and three Gram-negative bacterial strains by agar well diffusion. The results of the in vitro antibacterial screen of new compounds are shown in Table 1. For compound **9**, activity against four bacterial species was observed—*Micrococcus luteus, Staphylococcus aureus, Serratia marcescens*, and *Bacillus cereus* (Table 1)—whereas, for compounds **10a**, **10b**, and **10c**, no activity was observed against all bacterial species.

### 2.3. Antioxidant Activity

The antioxidant activity from organic compounds plays an important role through free radical scavenging, which is useful in the treatment of many diseases. In this study, the free radical scavenging activity of all compounds was carried out in the presence of the DPPH (1,1-diphenyl-2-picrylhydrazyl) and ABTS (2’-azino–bis(3-ethylbenzoline-6-sulfonic acid) diammonium salt) using ascorbic acid and α-Tocopherol antioxidant agents as a positive control. Although several methods are available for the determination of the antioxidant activity, the DPPH and ABTS methods are very common, rapid, and two of the most appropriate methods [14,15].

A series of concentrations ranging from 0.005–0.50 mg/mL were prepared and from each tested compound and then investigated for their antioxidant activity power using the different models. The results are summarized in Figure 1. The results indicated that the DPPH and ABTS radical scavenging activities were concentration dependent (Figure 1), and the order of radical scavenging power for both models was **9** > **10c** > **10b** > **10a**.

The IC50 (effective concentration for scavenging 50% of the inhibition) of synthesized compounds on DPPH and ABTS radicals are presented in Table 2. Based on the experimental results, among all the compounds synthesized **9**, **10a**, **10b**, and **10c** showed higher scavenging activity towards DPPH and ABTS. The higher antioxidant activity of compound **9** can be explained by the existence of the thiourea fragment [16]. However, the order of the radical scavenging power found in both models was **10c** > **10b** > **10a** due to the fact that the presence of the nitro group on the phenyl in compound **10c** determines a slight increase in antioxidant activity of compound **10c** compared with **10b** and **10a**.

## 3. Materials and Methods

### 3.1. Materials

2,4,6-trimethoxybenzaldehyde, hydroxylammonium chloride, dimethyl fumarate, hydrazine hydrate, substituted benzoyl chloride, phenyl isothiocyanate, 1,1-diphenyl-1-picrylhydrazyl (DPPH, purity N 99%), ascorbic acid (purity = 99%), α-tocopherol (purity = 99%), 2,2’-azino–bis(3-ethylbenzoline-6-sulfonic acid) diammonium salt (ABTS, purity N 99%), organic solvents and reagents were purchased from Aldrich (St Louis, MO, USA), Fluka (Steinheim, Germany) and Across (Morris Plains, NJ, USA) and were used without any further purification. Melting points were measured on electrothermal digital melting point apparatus and were uncorrected. Infrared spectra (IR) were recorded over the range 400–4000 cm^−1^ on an FT-IR Spectrometer Across (Thermo Scientific, WI, USA) and Janssen, Bruker spectrum 2000 Across (Morris Plains, NJ, USA). Potassium bromide pellets were used. High-resolution mass spectra (HRMS) were measured in positive ion mode using the electrospray ionization (ESI) technique on the Bruker APEX-2 instrument Across (Bremen, Germany) and Janssen. Nuclear magnetic resonance spectra (^1^H-NMR, ^13^C-NMR and 2DNMR) were measured using Bruker Avance III (Finnigan Corp., San Jose, CA, USA). Spectra were acquired in (CDCl_3_, 1% TMS) or (DMSO-*d*_6_, 1% TMS) 

### 3.2. Preparation of 2,4,6-Trimethoxybenzaldoxime

2,4,6-trimethoxybenzaldehyde (11.7g, 60 mmol) was dissolved in a 100 mL ethanol/NaOH (10%) mixture (1:1 ratio). Excess hydroxylammonium chloride was dissolved in water (75 mL). Then, the two solutions were mixed and the mixture was heated at 60 °C for 30 min. The mixture was then allowed to cool to room temperature. White crystals were collected by suction filtration and recrystallized from ethanol [17].

### 3.3. Preparation of 2,4,6-Trimethoxybenzonitrile Oxide ***1a***

2,4,6-trimethoxybenzaldoxime (4.8 g, 24 mmol) was dissolved in NaOH solution (1 N, 50 mL) and pyridine (20 mL). The clear solution was added dropwise with stirring for a period of 1 h to a previously prepared solution of Br_2_ (3.5 g) in ice-cooled 1 N NaOH (80 mL). The temperature was maintained during the addition at 0 °C. After the addition was completed, the resulting solution was stirred at 0 °C for a further 30 min. The resulting white precipitate was filtrated as quickly as possible through a large Büchner funnel, washed several times with ice-H_2_O, and dried under vacuum [18].

### 3.4. Preparation of trans-Dimethyl 3-(2,4,6-trimethoxy phenyl)-4,5-dihydro-4,5-isoxazoledicarboxylate ***5a***

Dimethyl fumarate (2.2 g, 15 mmol) was added to a solution of 2,4,6-trimethoxybenzonitrile oxide (3.15 g, 15 mmol) in dry tetrahydrofuran (THF) (80 mL). The mixture was heated under reflux for 6 h. The solvent was removed using a rotatory evaporator and the residue was recrystallized from the methanol-petroleum ether (60–80 °C) [19].

^1^H-NMR (400 MHz, CDCl_3_): δ ppm = 3.54 (s, 6H), 3.70 (s, 6H), 3.76 (s, 3H), 4.73 (d, 1H, *J* = 6.8 Hz), 5.45 (d, 1H, *J* = 6.8 Hz), 6.05 (s, 2H). ^13^C-NMR (100 MHz, CDCl_3_): δ ppm = 52.7, 52.9, 55.4, 56.0, 59.1, 80.5, 90.6, 98.0, 149.6, 159.7, 162.9, 167.8, 169.8.

### 3.5. Preparation of trans-3-(2,4,6-Trimethoxyphenyl)-4,5-dihydro-4,5-bis(hydrazenocarbonyl)isoxazole ***8a***

An excess amount of hydrazine hydrate was added to a solution of *trans*-dimethyl 3-(2,4,6-trimethoxyphenyl)-4,5-dihydro-4,5-isoxazoledicarboxylate (1.5 g, 4.3 mmol) in ethanol (80 mL). The mixture was heated under reflux for 50 h. The solvent was removed, and the residue was recrystallized from the methanol-petroleum ether (60–80 °C) to yield the product (1 g, 67% yield); mp: 178–180 °C. The HRMS displayed a molecular ion peak at *m*/*z* 376.12275 corresponding to the ion [C_14_H_19_N_5_O_6_ + Na]^+^ as expected from its calculated *m*/*z* = 376.12330. IR (KBr; cm^−1^): 3300 (N-H), 3050 (C-H arom), 2940 (C-H aliph), 1650 (C=O), 1600 (C=C), 1450 (C=N), 1100 (C-O).

^1^H-NMR (400MHz, DMSO-*d*_6_): δ ppm = 3.71 (s, 6H), 3.80 (s, 3H), 4.29 (bs, 4H, NH_2_), 4.50 (d, 1H, *J* = 6.8 Hz), 5.05 (d, 1H, *J* = 6.8 Hz), 6.23 (s, 2H), 9.14 (bs, 1H, NH), 9.50 (bs, 1H, NH). ^13^C-NMR (100 MHz, DMSO-*d*_6_): δ ppm = 55.4, 55.9, 58.7, 81.3, 90.9, 97.7, 150.5, 159.4, 162.2, 165.8, 166.9.

### 3.6. Preparation of trans-3-(2,4,6-Trimethoxyphenyl)4,5-dihydroisoxazolo-4,5-bis[carbonyl-(4′phenyl)thiosemicarbazide] ***9***

Phenyl isothiocyanate (0.82 g, 6 mmol) was added to a solution of hydrazinocarbonyl (3.1 g, 5 mmol) in absolute ethanol (30 mL). The reaction mixture was stirred at room temperature for approximately 20 h. The mixture was poured into cold water. The resulting precipitate was filtered and recrystallized from ethanol, affording 2.5 g (80%) of product **9**; mp: 192–194 °C, IR (KBr; cm^−1^): 3028-3363 (N-H), 1703 (C=O), 1200-1256 (C=S).

^1^H-NMR (400 MHz, DMSO-*d*_6_): δ ppm = 3.72 (s, 6H), 3.74 (s, 2H), 4.85 (d, 1H, J = 7.8 Hz), 5.40 (d, 1H, *J* = 7.8 Hz), 6.20 (s, 2H), 7.15-7.44 (m, 10H), 9.75 (bs, 4H, NH), 10.34 (bs, 1H, NH), 10.52 (bs, 1H, NH). ^13^C NMR (100MHz, DMSO-*d*_6_): δ ppm = 55.3, 56.0, 58.1, 81.6, 91.0, 97.3, 124.1, 125.2, 125.8, 128.0, 128.2, 138.6, 138.8, 139.0, 150.5, 159.4, 162.4, 180.3.

### 3.7. Preparation of trans-3-(2,4,6-Trimethoxyphenyl)-4,5-dihydroisoxazolo-4,5-bis(aroylcarbohydrazide) ***10a**–**c***

To a solution of acid dihydrazide (1.12 g, 2 mmol) and potassium carbonate (4 mmol) in aq. tetrahydrofuran (160 mL 1:3), a solution of benzoyl chloride (4 mmol) in tetrahydrofuran (20 mL) was added dropwise with stirring at 25 °C. The reaction mixture was stirred for 3–6 h, during which time a precipitate was formed. The precipitate was collected by filtration and washed with water, then recrystallized from dimethylsulfoxide (DMSO)/water (1:1).

For compound **10a**, *trans*-3-(2,4,6-trimethoxyphenyl)-4,5-dihydroisoxazolo-4,5-bis(phenylcarbohydrazide):

Yield: 1.48g (83%); mp: 202–204 °C, the HRMS displayed a molecular ion peak at *m*/*z* 560.17869 corresponding to the ion [C_28_ H_27_N_5_O_8_-H]^-^, as expected from its calculated *m*/*z* = 560.17814. IR (KBr; cm^−1^): 3145 (N-H), 3026 (C-H aromatic), 1675 (C=O), 1471 (C=N).

^1^H-NMR (400 MHz, DMSO-*d*_6_): δ ppm = 3.41 (s, 6H), 3.83 (s, 3H), 5.02 (d, 1H, *J* = 7.8 Hz), 5.31 (d, 1H, *J* = 7.8 Hz), 6.32 (s, 2H), 6.93-7.97 (m, 10H), 10.27 (bs, 1H, NH),10.44 (bs, 1H, NH), 10.59 (bs, 2H, NH). ^13^C-NMR (100 MHz, DMSO-*d*_6_): δ ppm = 55.4, 55.9, 58.2, 81.4, 91.0, 97.5, 124.9, 127.4, 127.5, 128.3, 128.5, 131.7, 131.9, 132.3, 132.4, 139.2, 150.2, 159.6, 162.3, 165.0, 165.2, 166.3, 167.4.

For compound **10b**, *trans*-3-(2,4,6-trimethoxyphenyl)-4,5-dihydroisoxazolo-4,5-bis(aroylcarbohydrazide)**:**

Yield: 0.86g (73%); mp: 225–227 °C. IR (KBr; cm^−1^): 3145 (N-H), 3026 (C-H aromatic), 1675 (C=O), 1471 (C=N).

^1^H-NMR (400 MHz, DMSO-*d*_6_): δ ppm = 3.76 (s, 6H), 3.79 (s, 3H), 3.83 (s, 6H), 4.97 (d, 1H, *J* = 7.8 Hz), 5.24 (d, 1H, *J* = 7.8 Hz), 6.27 (s, 2H), 7.11-7.50 (m, 8H), 10.22 (bs, 1H, NH),10.37 (bs, 1H, NH), 10.53 (bs, 2H, NH). ^13^C-NMR (100 MHz, DMSO-*d*_6_): δ ppm = 55.2, 55.3, 55.4, 55.9, 58.2, 81.5, 91.0, 97.5, 112.4, 112.4, 117.8, 119.7, 119.7, 129.5, 129.7, 133.7, 133.7, 150.2, 159.0, 159.2, 159.6, 162.3, 164.7, 163.9, 166.4, 167.3.

For compound **10c,**
*trans*-3-(2,4,6-trimethoxyphenyl)-4,5-dihydroisoxazolo-4,5-bis(aroylcarbohydrazide)**:**

Yield: 0.63g (52%) of product; mp: 255–257 °C, IR (KBr; cm^−1^): 3145 (N-H), 3026 (C-H aromatic), 1608 (C=O), 1471 (C=N), 1535 (NO_2_).

^1^H-NMR (400 MHz, DMSO-*d*_6_): δ ppm = 3.74 (s, 6H), 3.80 (s, 3H), 4.93 (d, 1H, *J* = 7.8 Hz), 5.28 (d, 1H, *J* = 7.8 Hz), 6.24 (s, 2H), 7.60-8.12 (m, 8H), 10.57 (bs, 1H, NH), 10.61 (bs, 1H, NH), 10.81 (bs, 2H, NH). ^13^C-NMR (100 MHz, DMSO-*d*_6_): δ = 55.3, 55.9, 58.2, 81.2, 90.8, 97.4, 124.2, 124.3, 129.5, 129.5, 130.1, 130.3, 131.3, 131.7, 146.9, 147.2, 150.2, 159.5, 162.3, 163.7, 164.0, 165.7, 166.8.

### 3.8. Antimicrobial Activity

In vitro antimicrobial activity of new compounds was screened against six different bacterial isolates (obtained from the Department of Biological Sciences, Yarmouk University, Jordan) using the agar well diffusion methods. The six bacterial isolates investigated included three Gram-positive bacteria *Micrococcus luteus, Bacillus cereus*, and *Salmonella* typhi and three Gram-negative bacteria *Staphylococcus aureus, Serratia marcescens,* and *Pseudomonas aeruginosa*. Bacterial strains were cultured overnight at 37 °C in tryptone soy broth (TSA).

### 3.9. Antioxidant Activity

The antioxidant activity of the compound was determined using the DPPH and ABTS according to the procedures described in the literature [19,20,21,22]. Positive controls used included α-tocopherol and ascorbic acid, while methanol was the negative control. All determinations of the IC_50_ by the three assay methods were conducted in triplicate. The IC_50_ of the extracts and the positive controls, expressed as mean ± SD, are shown in Table 2. All determinations of the IC_50_ by the three assay methods were conducted in triplicate.

## 4. Conclusions

New 2-isoxazoline derivatives were successfully synthesized and characterized using spectroscopic techniques (IR and NMR) and elemental analysis. All the synthesized compounds have been investigated for their antioxidant activity by DPPH and ABTS assays, and the results indicated that these compounds have good scavenging activities. They were evaluated for their antimicrobial activities against certain Gram-positive and Gram-negative bacteria. The results show that compound 9 has activity against certain bacterial species, whereas the remainder of the compounds have no considerable effects on microbial growth.

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
