# Peer review of "Synthesis, Antimicrobial and Antioxidant Activities of 2-Isoxazoline Derivatives"

_molecules, 2020, doi:10.3390/molecules25184271_

Round 1

Reviewer 1 Report

The manuscript entitled “Synthesis, antimicrobial and antioxidant activities of 2-isoxazoline derivatives” demonstrated that comparison in between antimicrobial activity and antioxidant activity of various 2-isoxazoline derivatives. The data and methods of this paper is not proper; the research data to support results and conclusion is weak in the current form. The issues are listed as below.

  1. Introduction is very weak, no clear motivation details Only mentioned in 1 paragraph.
  2. Antimicrobial activity must include graph and images of bacterial cell inhibition nt just a table without any validation.
  3. Antioxidant activity also need graph at various concentration not just table.
  4. Author must present strong evidence for their presented concept of antimicrobial activity of 2-isoxazolines. Author must provide experimental results related molecular mechanism and microscopic images for in-depth study (such as PCR, blots and microscopic images)
  5. Author must do toxicity evaluation on cells or cell lines. Toxicity of any physical and chemical agent must be evaluated on primary or normal cells for validation.
  6. Quality of data is weak and not validated with strong proof, author must improve quality of paper by validation using real images, and spectra’s.

Author Response

I thank the Reviewers for their valuable opinions, and I modified and corrected most of the required, but for antimicrobial activity, I did not measure MIC or MBC value and did not take pictures for molecular mechanism and microscopic images for in-depth study (such as PCR, blots and microscopic images). The spectra (IR, NMR) of compounds are added as a supplementary file. Whereas for antioxidant the Figure 1 at various concentration Has been added.

  1. Introduction is very weak, no clear motivation details Only mentioned in 1 paragraph.

Has been Modified

  1. Antimicrobial activity must include graph and images of bacterial cell inhibition nt just a table without any validation.

Only I made screening for antimicrobial, the MIC or MBC value I could not measure so that I corrected the caption of table 1

  1. Antioxidant activity also need graph at various concentration not just table.

Figure 1 at various concentration for antioxidant Has been added.

  1. Author must present strong evidence for their presented concept of antimicrobial activity of 2-isoxazolines. Author must provide experimental results related molecular mechanism and microscopic images for in-depth study (such as PCR, blots and microscopic images).

Only I made screening for antimicrobial, the MIC or MBC value I could not measure so that I corrected the caption of table 1

  1. Author must do toxicity evaluation on cells or cell lines. Toxicity of any physical and chemical agent must be evaluated on primary or normal cells for validation.

This is another work, Only I made screening for antimicrobial

  1. Quality of data is weak and not validated with strong proof, author must improve quality of paper by validation using real images, and spectra’s.

The spectra of compounds are added as a supplementary file.

Reviewer 2 Report

Alshamari and his/her colleagues (manuscript ID 922393) present a study of drugs for antimicrobial or antioxidant activities. The overall study is technical sound and provides an excellent candidate for further antimicrobial or antioxidant activities. I only have a few minor comments: 1. Salmonella Typhi is not a species, it is a serovar named Typhi, so only Salmonella is italic. 2. There are very bacterial strains that were tested in the current version and an expanded list of bacteria that should be included in further evaluation. Additionally, the MIC or MBC value for individual drugs which has good candidacy is needed. 3. It is not know how the drug synthesis experiment is replicated, that is to say, detailed material and method is needed for the parameter of synthesis of different variants. 4. A detailed introduction and discussion, with the most updated references, for a rational study desgin is needed.

Author Response

I thank the Reviewers for their valuable opinions, and I modified and corrected most of the required, but for antimicrobial activity, I did not measure MIC or MBC value and did not take pictures for molecular mechanism and microscopic images for in-depth study (such as PCR, blots and microscopic images). The spectra (IR, NMR) of compounds are added as a supplementary file. Whereas for antioxidant the Figure 1 at various concentration Has been added.

  1. Salmonella Typhi is not a species, it is a serovar named Typhi, so only Salmonella is italic.

I corrected it

  1. There are very bacterial strains that were tested in the current version and an expanded list of bacteria that should be included in further evaluation. Additionally, the MIC or MBC value for individual drugs which has good candidacy is needed.

Only I made screening for antimicrobial, the MIC or MBC value I could not measure so that I corrected the caption of Table 1>

  1. It is not know how the drug synthesis experiment is replicated, that is to say, detailed material and method is needed for the parameter of synthesis of different variants.

I have modified it

  1. A detailed introduction and discussion, with the most updated references, for a rational study design is needed.

I am added two references

Round 2

Reviewer 1 Report

I again ask authors to revise manuscript according to my previous comments on biological activities. Author can provide in silico data, If author can’t do new experiments laboratory due to covid19 situation. I recommend author to revise manuscript again.

also English language is very poor, it must be edited by professional english editing company. Author can choose MDPI English editing services or any other editing company.

Author Response

dear reviewer 

I would to thank you 

The English language was editing by MDPI English editing services.

The biological results are present in the manuscript, it is the only one that I have, and it has been fully included in the manuscript, and I apologize that I did not finish the work due to the covid19 condition.